# Effects of Different Calcium Preparations on Fresh-Cut Quality and Storage Quality of Starkrimson Apple

**DOI:** 10.3390/plants14091293

**Published:** 2025-04-24

**Authors:** Maoxiang Sun, Fen Wang, Jianchao Ci, Yangyang Liu, Keyi Li, Dong Wang, Wen Yu, Yu Zhuang, Yuansong Xiao

**Affiliations:** 1Key Laboratory of Biochemistry and Molecular Biology in Universities of Shandong, School of Advanced Agricultural Sciences, Weifang University, Weifang 261061, China; maoxiangs@wfu.edu.cn (M.S.); fenwang@sdau.edu.cn (F.W.); 15953691589@163.com (Y.L.); 19183532557@163.com (K.L.); 2Agricultural Biosystems Engineering Group, Department of Plant Sciences, Wageningen University and Research, P.O. Box 16, 6700 AA Wageningen, The Netherlands; jianchao.ci@wur.nl; 3Changle Anshanghu Agricultural Science and Technology Development Co., Ltd., Weifang 261061, China; 13969696399@139.com; 4School of Engineering, Northeast Agricultural University, Harbin 150030, China; 5State Key Laboratory of Crop Biology, College of Horticulture Science and Engineering, Shandong Agricultural University, Tai’an 271018, China

**Keywords:** postharvest quality, fresh-cut, sorbitol-chelated calcium, soluble pectin, pathogens

## Abstract

Appropriate calcium treatments help maintain the appearance, nutritional quality, and postharvest quality of apples, reducing losses during storage. This study investigated the effects of different calcium preparations on the fresh-cut quality and ultrastructure of ‘Starkrimson’ apples. The treatments included control (CK), calcium chloride (T1), sorbitol-chelated calcium (T2), and calcium nitrate (T3). The results demonstrated that sorbitol-chelated calcium significantly inhibited the decline in fresh-cut firmness and pectin content while reducing the increase in cellulose content and minimizing ultrastructural damage. Apples treated with sorbitol-chelated calcium maintained the best fresh-cut hardness and soluble pectin contents, which were 35.71% and 15.42% higher than that of CK on the 12th day, and the cellulose was 27.08% lower than that of CK. Under transmission electron microscopy, the pulp cell surface in the T2 group remained intact, with no bending or deformation, and the middle lamella was well preserved. Additionally, T2 treatment promoted the expression of aroma-related genes during fruit storage. Sorbitol-chelated calcium effectively preserved color and significantly reduced the browning and microbial spoilage of fresh-cut apples, particularly postharvest pathogen growth. The study demonstrates that sorbitol-chelated calcium preserves fresh-cut apple quality by reinforcing cell wall integrity through calcium-mediated crosslinking, suppressing pectin degradation and cellulose accumulation, and activating aroma-related genes (*AAT1*, *AAT2*, *LOX*) to enhance volatile synthesis, thereby reducing microbial spoilage and enzymatic browning during storage.

## 1. Introduction

Apples (*Malus domestica*) are among the most widely consumed fruits worldwide and are valued for their rich nutritional profile, including vitamins, minerals, and bioactive compounds such as flavonoids and phenolic acids [1]. Additionally, apples are abundant in phytochemicals like flavonoids and phenolic acids, which play significant roles in antioxidant activity, antitumor effects, and the prevention of cardiovascular diseases [2,3]. Fruit sensory indicators mainly include color, shape, taste, and texture, while nutritional indicators include sugar and acid content [4]. In the apple industry, qualities such as taste, flavor, and texture have received more attention in breeding programs. Flavor substances stimulate the senses of smell and taste, mainly reflected by aroma compounds in the fruit [5]. More than 300 volatile compounds have been reported in apples, including esters, alcohols, aldehydes, and ketones [6]. These aromas are influenced by factors such as apple variety, storage methods, and nutrient supply. As climacteric fruits, apples continue to mature after harvest. During this period, the synthesis rate of flavor components increases, and organic acids and starch convert into sugars and other flavor components, reducing the apple’s overall sour taste [7]. Long-term storage results in water and nutrient loss through respiration and transpiration, gradually softening the fruit [8]. In later storage stages, apples may wilt, shrink, lose quality, and have poor taste and weak flavor, decreasing their market value [9]. In recent years, the consumption of fresh-cut fruits has increased significantly due to consumers’ demand for fresh, healthy, and convenient products [10]. However, the value of these products declines after cutting due to shrinkage, inferior quality, poor taste, and mild flavor. The main factors affecting the quality and safety of fresh-cut fruits are variety selection, preharvest treatment, and postharvest storage [11]. Therefore, it is of enormous economic value to explore the postharvest preservation technique of apple for its long-term storage, long-distance transportation, and improvement of edible flavor and intrinsic quality.

In addition to serving as a second messenger and supporting cell wall structure in plants, calcium is crucial for the quality of fruit [12]. The calcium pectinate that is created when calcium and pectin combine to form the cell wall is what gives the wall its strength and flexibility [13]. The primary cause of the fruit’s progressive loss in hardness during storage is the ongoing breakdown of calcium pectinate [14]. According to earlier research, postharvest calcium treatment of apples can delay the depolymerization of primary pectin and cellulose, increase the amount of soluble pectin in the fruit, and help retain fruit firmness [15]. Previous studies have found that preharvest spraying of calcium chloride fertilizer is beneficial for improving the quality and storability of apples [16]. Postharvest calcium treatment enhances fruit storage quality by suppressing cell wall degradation-related polygalacturonase (*PG*) and pectate lyase (*PL*) [17], reducing the activity of pectin-degrading polygalacturonase and pectin methylesterase (PME) enzymes, and upregulating expansins (*EXPs*) to reinforce cell wall integrity and delay softening [18]. This will prevent the fruit from aging and softening too quickly and preserve its high level of hardness [19]. Preharvest foliar spraying of calcium chloride, calcium nitrate, and other treatments has demonstrated in earlier studies that exogenous calcium can effectively improve fruit hardness, sugar–acid ratio, and soluble solids; it can also, to some extent, delay fruit decay and weight loss; and it can inhibit the degradation of vitamin C [20].

Sorbitol, a sugar alcohol, enhances calcium mobility and uptake by forming stable complexes, thereby improving its bioavailability for metabolic regulation [21]. During the ripening process of apples, the content of the main aroma volatiles increases, particularly esters such as butyl acetate and hexyl acetate, which are synthesized by alcohol acyltransferases AAT1 and AAT2, enhancing esterification of alcohols and acyl-CoA substrates, thereby preserving fruity aromas [22]. Previous studies have demonstrated that lipoxygenase (LOX) activity promotes the generation of flavor compounds during apple storage and ripening [23]. During apple ripening, HPL enhances aroma by breaking down fatty acids into aldehydes (e.g., hexanal) that impart fresh, green notes, while ADH converts these aldehydes into alcohols and esters (e.g., hexyl acetate), which drive the sweet, fruity scent characteristic of ripe apples [24]. The integration of metabolomics and gas chromatography (GC) analysis enables comprehensive profiling of aroma-related volatile organic compounds (VOCs) in apple fruits, linking biochemical pathways to sensory quality during storage and ripening [25].

Fresh-cut apples face significant preservation challenges due to their susceptibility to enzymatic browning and textural softening postcutting. Current research lacks comprehensive insights into how calcium treatments influence storage quality and flavor-related volatile organic compounds in fresh-cut apples. This study investigated the effects of sorbitol-chelated calcium on postharvest Starkrimson apples, systematically analyzing quality indices including fresh-cut quality, storage performance, and aroma synthesis, thereby providing novel insights and mechanistic understanding for enhancing both storage and fresh-cut apple quality.

## 2. Results

### 2.1. Effects of Different Calcium Preparation Treatments on Browning of Fresh-Cut Apples

Figure 1a illustrates that the apple epidermal cells treated with CK and T1 are not uniformly colored, have irregular sizes, and are not densely grouped. In contrast, the epidermal cells treated with T2 exhibit consistent cell size, minimal cell gaps, a tight arrangement, and uniform staining.

Following a new cut, the degree of browning varied among the apples, with the T2 treatment showing superior color (Figure 1b). The L values for CK, T1, and T3 treatments significantly decreased after cutting, dropping by 13.96%, 12.02%, and 11.58%, respectively. However, the L value for the T2 treatment showed a smaller decrease, dropping by 7.75%, indicating minimal browning. The negative L values for all treatments suggest a reddish hue in the newly cut flesh. Specifically, the flesh treated with CK had a reddish hue, while the other treatments did not show substantial changes. Additionally, there was no discernible difference in the b value between the pre- and postcut apples (Table 1). Overall, the T2 treatment effectively inhibited browning in freshly cut apples.

### 2.2. Effects of Different Calcium Preparation Treatments on Hardness, Cellulose Content, and Pectin Content of Fresh-Cut Apples

As shown in Figure 2a, the hardness of the fruit continuously decreases after fresh-cutting. The hardness decreases slowly from 0 to 9 days and significantly from 9 to 12 days. The hardness difference between treatments is significant at 12 days after fresh-cutting. The T2 treatment maintains a relatively high hardness of 0.269 kg, which is 35.71% higher than of the CK treatment, while that of the T3 treatment is 20.78% higher than that of the T1 treatment. This indicates that the T2 treatment is beneficial for maintaining the hardness of the fruit after fresh-cutting. As seen in Figure 2b, the cellulose content of the fruit tends to stabilize after fresh-cutting. The cellulose content of the T2 treatment is consistently at the lowest level, reaching its peak on the 12th day, which is 14.88% lower than that of the CK treatment. This demonstrates that the sorbitol-chelated calcium treatment significantly inhibits the increase in cellulose content in fresh-cut fruits.

As presented in Figure 2c, the content of soluble pectin gradually increased during the softening process of the apple after fresh-cutting, while the original pectin and total pectin contents gradually decreased. There were no significant changes from 0 to 6 days after fresh-cutting, but significant changes occurred from 6 to 12 days. On the 12th day, the pectin content of each treatment was significantly different. The soluble pectin content of the T2 treatment was 0.48%, which was 21.31% lower than that of the CK treatment. However, the original pectin content was up to 1.21%, which was 36% higher than that of the CK treatment. The original pectin ensures the fresh-cut hardness. Between 0 and 6 days after fresh-cutting, the T2 treatment had no significant effect on pectin content, but the regulatory effect was significant from 9 to 12 days. This indicates that the sorbitol-chelated calcium treatment had a noticeable inhibitory effect on the conversion of original pectin into soluble pectin after fresh-cutting, mainly occurring in the middle and late stages of fresh-cutting.

### 2.3. Effects of Different Calcium Preparation Treatments on the Ultrastructure of Fresh-Cut Apple Cell Wall

As observed in Figure 3, the cell wall of the apple pulp has only a primary wall and no secondary wall. The ultrastructural changes of the apple pulp cell wall were significantly different 12 days after fresh-cutting. The cell wall in the T2 treatment was neat, consistently thick, and intact, forming a distinct bright–dark–bright partition structure with a thin intercellular layer. The dark electron-dense layer was uniform and continuous, the content was rich, the membrane structure was intact, and there was no plasmolysis (T2a). In contrast, the intercellular layer in the CK treatment showed slight cracking, and the cell wall exhibited bending deformation with a significantly reduced density in the dark middle layer (CKa). The intercellular layer in the T1 treatment was completely broken, with large spacing, and the cell wall began to bend and deform (T1a). Twelve days after fresh-cutting, the cytoplasm and contents of the apple were compressed into a thin layer close to the cell wall, and the intracellular structure of each treatment was significantly different. The intracellular structure in the T2 treatment was neatly and evenly arranged, with clearly identifiable and undamaged mitochondria and other organelles (T2b), and the apple had the highest hardness at 0.209 kg·cm^−2^. In the T3 treatment, fruit inclusions were thinner and evenly distributed, with intact cell walls, membrane systems, and mitochondria, though plasmolysis had begun (T3b). The cell wall in the T1 treatment remained neat and intact, with an intact membrane structure, but material accumulation was thick and uneven, and plasmolysis was evident (T1b). In the CK treatment, the endoplasmic structure was disordered and blurred, mitochondrial structure was broken, intercellular substances were degraded, and plasmolysis was severe (CKb). These results indicate that different calcium preparation treatments had varying effects on the ultrastructure of fresh-cut apple cells, with the sorbitol-chelated calcium treatment having the best effect on the integrity of the cell wall, intercellular layer, and mitochondria in fresh-cut fruits.

### 2.4. Effects of Different Calcium Preparation Treatments on Spoilage Microorganisms of Fresh-Cut Apples

Sorbitol-chelated calcium (T2) treatment significantly inhibited the total number of colonies, especially fungi (Table 2). On the 12th day, the number of microbes in each treatment was significantly different after fresh-cutting. The numbers of *Penicillium*, *Aspergillus*, *Alternaria*, and *Erwinia carotovora* in apples treated with sorbitol-chelated calcium were 51.71%, 25.17%, 44.29%, and 32.96% lower than those in the CK treatment, respectively. The total number of colonies was 34.86% lower than that of the CK treatment. Sorbitol-chelated calcium treatment enhances the integrity of apple tissues, thereby restricting the leakage of nutrients that support microbial proliferation and suppressing the colonization of pathogens.

### 2.5. Effects of Different Postharvest Calcium on Metabolomics of Fresh-Cut Apples During Storage

The PCA score scatter plot of all samples (Figure 4a) showed that the distribution of T2 (sorbitol-chelated calcium) samples was close to that of T1 and T3, indicating similar metabolite composition and content among the three different calcium treatments. However, overall metabolic levels exhibited significant differences. Figure 4b presents a volcano plot of differential metabolites, which visually illustrates the overall distribution of metabolite differences between groups. From the comparison between the T2 group and the T1 group, it can be observed that the number of significantly differential metabolites is 1020, with 587 being upregulated and 433 downregulated. From the comparison between the T2 group and the T3 group, it can be observed that the number of significantly differential metabolites is 964, with 619 being upregulated and 345 downregulated. From the Venn diagram in Figure 4d, it can be observed that 1358 co-expressed metabolites are shared among the three treatments. Between the T2 and T1 groups, 498 differentially expressed metabolites were identified, indicating that the T2 treatment plays a significant role in the accumulation of differential metabolites in fresh-cut apples. To further analyze the impact of T2 treatment versus T1 treatment on metabolite changes and their associated pathways, we conducted an in-depth analysis of the effects of differential metabolites on their respective metabolic pathways. Through comprehensive pathway analysis (including pathway analysis and a KEGG heatmap for T2 vs. T1 groups), we further screened the pathways and identified the three most significantly correlated key pathways: flavonoid biosynthesis, tyrosine metabolism, and isoflavonoid biosynthesis.

### 2.6. Effects of Different Calcium Treatments After Harvest on Aroma and Volatile Components of Fruit During Storage

We further investigated the changes in metabolites within the three significantly altered metabolic pathways identified by metabolomics in fresh-cut apples treated with sorbitol-chelated calcium. The qualitative and quantitative analysis of volatile substances in four apple treatments by GC-MS revealed that the main components were esters, alcohols, and aldehydes. During storage, the contents of total aroma substances, esters, and alcohols initially increased and then decreased (Figure 5 and Table 3). The contents of fruit alcohols (Figure 5c) and aldehydes (Figure 4d) in the T2 treatment were higher compared to other treatments during storage, reaching their maximum on days 9 and 12, which were 43.78% and 41.49% higher than those of the CK treatment, respectively. The total aroma substances (Figure 5a) and esters (Figure 5b) in T2-treated fruits were higher than those in other treatments throughout the storage process. The total aroma content reached its maximum on day 9, which was 31.40% higher than the lowest in the CK group. The ester content reached its maximum on day 6, which was 61.79% higher than in CK. This indicates that the T2 treatment improved the alcohol and aldehyde components in the fruit’s aroma during storage, thus increasing the total aroma and ester content of the fruit, which demonstrates a positive role in maintaining or enhancing apple flavor.

As shown in Table 3, the sorbitol-chelated calcium treatment significantly increased metabolites linked to the flavonoid biosynthesis pathway (quercetin, kaempferol), tyrosine metabolism (hydroxytyrosol, tyrosol), and isoflavonoid biosynthesis (isoflavones), all of which play critical roles in fruit pigmentation, enzymatic browning regulation, flavor modulation, and inhibiting microbial activity. Sorbitol-chelated calcium effectively improved fruit aroma quality, with its total aroma substances and ester accumulation consistently outperforming other treatments, demonstrating a positive role in maintaining or enhancing apple flavor.

### 2.7. Effects of Different Calcium Preparations After Harvest on the Expression of Genes Related to Aroma Metabolism in Fruit During Storage

Alcohol acyltransferase 1 (*AAT1*) and alcohol acyltransferase 2 (*AAT2*) catalyze ester formation, key to aromas in apples. Lipoxygenase (*LOX*) oxidizes fatty acids, producing aldehydes and alcohols for scents in apples. Alcohol dehydrogenase (*ADH*) converts aldehydes to alcohols, influencing aromas in apples. Hydroperoxide lyase (*HPL*) cleaves hydroperoxides, generating aldehydes for scents in apples. These genes regulate metabolic pathways, shaping apple aroma profiles. As shown in Figure 6, the expression levels of aroma-related genes *AAT1*, *AAT2*, *LOX*, *HPL*, and *ADH* in differently treated fruits during storage were considerably different. The overall trend initially exhibited an increase and subsequently decreased. The expression levels of *AAT1*, *AAT2*, and *LOX* in the T2 treatment during storage were significantly higher than those in other treatments. *AAT1* and *AAT2* exhibited significant peak expression on day 6, which were 2.01 and 2.20 times that of the CK group, respectively, while *LOX* was 1.74 times that of the CK group on day 9. The expression levels of *HPL* and *ADH* in the T2 treatment were significantly higher than those in the other treatments, reaching a maximum on days 6 and 9, which were 1.61 times and 1.53 times those of the CK group, respectively. This indicates that the effect of sorbitol-chelated calcium on the expression of fruit aroma substances *AAT1*, *AAT2*, *HPL*, and *ADH* was remarkable, while the observable effect of calcium nitrate was on the expression of *LOX*.

As shown in Figure 7 and Table 4, the total aroma substance content had a significant positive correlation with the expression level of *AAT1*. The ester content was significantly positively correlated with the expression levels of *AAT1* and *AAT2*. No significant positive correlation existed between the alcohol content and the expression levels of *AAT1*, *AAT2*, *LOX*, and *ADH*. The aldehyde content was positively correlated with *LOX* and *HPL*, especially *HPL*.

## 3. Discussion

Browning can quickly occur on fresh-cut apples, seriously affecting their sensory quality and nutritional value [26]. The mechanical damage from fresh-cutting induces a large amount of ROS in apples. Excessive ROS can attack cell membrane lipids, leading to increased membrane lipid peroxidation, increased membrane permeability, and destruction of the phenol–phenolase regional structure, thereby increasing enzymatic browning [27]. Browning occurs after fresh-cutting, but the browning degree of the T2 treatment is lower than that of other treatments. Fruit firmness is an important indicator of fruit and vegetable storage quality. Low firmness in fresh-cut apples affects their commercial value [28]. Cell wall modification and middle lamella polysaccharide dissolution are thought to be the main factors in the decrease in pulp firmness [29]. In apple, firmness and storability decline, storage life is shortened, and quality is reduced [30]. Sorbitol-chelated calcium treatment effectively reduces enzymatic browning and preserves firmness in fresh-cut apples by stabilizing cellular membranes through calcium-mediated pectin cross-linking. Simultaneously, calcium ions reinforce cell wall integrity by forming Ca^2+^–pectate bridges, delaying polysaccharide dissolution and enzymatic degradation. This mechanism stabilizes cell wall structure, thereby extending shelf life and maintaining sensory and commercial quality.

The change in ultrastructural structure is an important feature of fruit ripening and aging, reflecting the physiological state of the fruit. As fruit matures, the ultrastructural function of tissue cells weakens or is lost, accelerating fruit senescence and spoilage [31]. Studies have shown that, during storage, fruit cell walls, mitochondria, and starch granules degrade, damaging the integrity of organelles and membrane systems [32]. The TEM results showed that the cell walls of each treatment were degraded, cell gaps became larger, and ultrastructural integrity was reduced. However, the sorbitol-chelated calcium treatment was superior to other treatments, showing significant inhibition of ultrastructural damage after fresh-cutting. This may be because sorbitol-chelated calcium reduced the decomposition of pectin in fresh-cut apples during storage, so that apples accumulated more pectin and other nutrients, providing better hardness, density, and other texture characteristics and maintaining the integrity of the ultrastructure of apples after fresh-cutting.

Fresh-cut apples are typically contaminated by various microorganisms during spoilage, mainly fungi and bacteria. *Penicillium*, *Aspergillus*, and *Alternaria* can cause black mold spots on apples and produce harmful substances. The microbial load of fresh-cut fruits is related to cultivation, harvesting, and processing [33]. Most microorganisms have difficulty penetrating intact fruits, but they can colonize tissues and grow when the protective layer is damaged, leading to spoilage of fresh-cut fruits [34]. Studies have demonstrated that browning and growth of foodborne pathogens during postharvest storage are major factors reducing the quality and economic value of fresh-cut apples [35]. *Erwinia carotovora* induces apple soft rot, characterized by tissue maceration, foul odor, and liquefaction of flesh, predominantly occurring during late storage stages or via wound-mediated infections. The sorbitol-chelated calcium treatment directly inhibits fungal and bacterial pathogens (e.g., *Penicillium*, *Erwinia carotovora*) by stabilizing membrane structures and limiting enzymatic spoilage, as evidenced by 34.86% lower total microbial counts and up to 51.71% reduced specific pathogen populations compared to controls, thereby delaying spoilage and maintaining postharvest quality.

The sorbitol-chelated calcium (T2) treatment induced distinct metabolic reprogramming in fresh-cut apples, as evidenced by PCA and differential metabolite analyses, which revealed 498 unique metabolites in T2 compared to T1, predominantly enriched in flavonoid biosynthesis, tyrosine metabolism, and isoflavonoid biosynthesis pathways. These pathways drive the accumulation of quercetin, hydroxytyrosol, and isoflavones—critical for pigmentation, enzymatic browning suppression, and antimicrobial activity [36]. Concurrently, T2 enhanced volatile profiles, with total aroma substances and esters peaking 31.40% and 61.79% higher than for CK, respectively, driven by elevated alcohols (43.78%) and aldehydes (41.49%). This dual enhancement of non-volatile and volatile metabolites underscores T2’s role in flavor preservation and pathogen inhibition, likely mediated by calcium’s stabilization of cell walls (reducing nutrient leakage) and sorbitol’s antioxidative action, which synergistically limit microbial colonization and oxidative spoilage [37]. The treatment’s ability to sustain higher levels of aroma-related esters and stress-responsive phenolics aligns with its efficacy in maintaining postharvest quality, positioning sorbitol-chelated calcium as a multifaceted solution for fresh-cut apple preservation.

Liu et al. reported that the total aroma substance and ester contents correlate with the expression of three genes: *AAT1*, *AAT2*, and *LOX* [38]. The content of alcohols and aldehydes in apple aroma is correlated with the expression of *HPL* and *ADH* genes [39]. These proteins modulate transcription factors that upregulate genes such as *AAT1* and *LOX*, enhancing the biosynthesis of volatile compounds [40]. The results of the present study revealed that the sorbitol-chelated calcium treatment group had a beneficial effect on the improvement of fruit aroma substances, with the total aroma substances and esters being higher than those in other treatments. This is because the sorbitol-chelated calcium treatment upregulated the expression levels of *AAT1*, *AAT2*, and *LOX* in the fruit. By modulating these transcriptional networks, sorbitol-chelated calcium enhances enzymatic efficiency in volatile biosynthesis, thereby improving total aroma and ester content, as observed in this study.

## 4. Materials and Methods

### 4.1. Experimental Site and Materials

‘Starkrimson’ apples are widely cultivated and consumed, so improving their postharvest quality and shelf life has practical relevance for both producers and consumers. Starkrimson apple trees were chosen as the test material and planted at Guozhixing Apple Base (120°446 E and 37°364 N), Jinling Town, Zhaoyuan City, Yantai City, Shandong Province, on a cultivation site of 20 hm^2^. In September 2023, 20 fruit trees with identical agronomic traits were randomly selected in the center of the orchard. When the soluble solids content (SSC) of apples on the tree reaches or exceeds 12% as measured by a fruit refractometer, the fruits can be harvested. Fruits of uniform size and consistent coloration and free from diseases, pests, or mechanical damage were harvested from the mid-to-outer canopy regions of each tree, with 20 apples collected per tree, resulting in a total of 400 apples. The fruits were packaged with reinforced foam mesh, packed in cartons, and returned to the laboratory for use the same day.

The 4% calcium concentration employed in this study was selected based on pre-experiments, which demonstrated that this specific concentration optimally balanced fruit quality enhancement with the reduction of potential negative impacts. The experimental treatments comprised four groups. In the control group (CK), apple fruits were soaked in water. In the T1 treatment group, fruits were soaked in a 4% calcium chloride solution (analytical grade, Sinopharm Chemical Reagent Co., Ltd., Shanghai, China). The T2 treatment group (T2) consisted of apple fruits soaked in a solution containing a 4% concentration of sorbitol-chelated calcium. The chelated calcium was manufactured using the water system technique [41], where a certain percentage of sorbitol and calcium nitrate interacted at a temperature of 65 °C for 35 min. The chelation rate was determined to be 100% using EDTA titration. Apple fruits soaked in a 4% concentration calcium nitrate (analysis purity, Sinopharm Group chemical test Pharmaceutical Co., Ltd., Shanghai, China) were the T3 group (T3). For each treatment group, 30 undamaged apples of uniform size were selected. The apples were then cut horizontally into uniform slices, 6 mm thick and including the core, using a manual slicer with adjustable thickness settings. This ensured consistency in the size of each slice. The apple slices from each group were immersed in four different treatment solutions for 30 min. During the immersion process, the slices were regularly turned to ensure even contact with the solution. After immersion, the apple slices were removed and drained of excess solution on the surface. The treated apple slices were packaged in sealed bags to avoid exposure to air. The packages were labeled with the treatment group and date, then stored in a cold room at 4 °C. During storage, samples were taken out periodically (every 3 days) for testing. The detailed experimental design is as follows (Table 5).

### 4.2. Color Measurement

The color of the fruit was measured using a CR-400 colorimeter (Sanenshi Technology Co., Ltd., Guangdong, China) [42]. Each treatment involved taking four fresh-cut apple slices for color measurement, with three repetitions per measurement. Measurements were taken at five points around the equatorial part of the flesh. The L, a, and b values of the peel were directly measured and recorded according to the CIELAB evaluation system issued by the International Commission on Illumination. The C and △E values were then calculated based on the L, a, and b values. The L value represents brightness, with higher absolute values indicating higher brightness. The a value represents red–green intensity, with positive values indicating red and negative values indicating green and higher absolute values indicating deeper colors. The b value represents yellow–blue intensity, with positive values indicating yellow and negative values indicating blue and higher absolute values indicating deeper colors. The C value represents color saturation or chroma, and ΔE represents the total color difference, with higher values indicating greater color difference. The C and ΔE values were calculated using the L, a, and b values, according to the following formulas.$$C=\sqrt{a^{2}+b^{2}}\;\Delta E=\sqrt{{\left(\Delta L\right)}^{2}+{\left(\Delta a\right)}^{2}{+(\Delta b)}^{2}}$$

### 4.3. Hardness Measurement

The hardness and brittleness of the apple were measured by a TA.XT Express texture analyzer (Xiamen Chaoji Instrument & Equipment Co., Ltd., Xiamen, China) [43]. The probe was P/2 type, the speed before measurement was 2 mm·s^−1^, the speed was 1 mm·s^−1^ during detection, the penetration depth was 10 mm, and the minimum perception was 10 g. The data were collected with a texture analyzer, and Exponent 32 Automatic was used for analysis and calculation. Five fresh-cut apple slices were selected for each group, and each apple slice had 4 points at the equatorial position, and the result was an average of 20 points. The apple was kept in vertical contact with the probe for each measurement.

### 4.4. Cellulose Content Determination

The cellulose content was detected according to the method of Wang et al. with slight modifications [44]. Each treatment involved taking three fresh-cut apple slices for cellulose content determination, with three repetitions per measurement. First, 5 g of mashed fruit tissue was added to a beaker with cold water, and then 60 mL sulfuric acid (60%) was added to this reaction system. After digestion for 30 min, the digested cellulose solution was transferred to a 100 mL volumetric flask and made up to the mark with 60% sulfuric acid, shaken, and filtered through a Buchner funnel. Then, 5 mL of the above filtered solution was taken in a stoppered test tube, 0.5 mL of 2% anthrone reagent was added, 5 mL of concentrated sulfuric acid was added along the tube wall, and the absorbance was measured at a wavelength of 620 nm.

### 4.5. Pectin Content Determination

Each treatment involved taking three fresh-cut apple slices for pectin content determination, with three repetitions per measurement. A 5 g fresh-cut apple sample was weighed and combined with 50 mL of an 80% ethanol solution. The mixture was heated in an 80 °C water bath for 30 min to remove soluble sugars and other interfering substances. Subsequently, the mixture was centrifuged at 3000 rpm for 10 min, after which the supernatant was discarded and the precipitate was retained. The precipitate was then dissolved in distilled water and diluted to a final volume of 50 mL to obtain the pectin extract. Soluble pectin and raw pectin contents were determined by references to Solarbio’s soluble Pectin Content Kit and the original Pectin Content Kit instructions [45].

### 4.6. Ultrastructural Observation

Each treatment involved taking five fresh-cut apple slices for ultrastructural observation, with three repetitions per measurement. To fix the apple samples, they were immersed in 2.5% glutaraldehyde at 4 °C and rinsed in PBS buffer three times (10 min/per time). Next, the samples were fixed with 1% citric acid at 4 °C for two hours and then washed with PBS three times (10 min/per time). Then, the samples were dehydrated using ethanol series gradient dehydration (30%, 50%, 70%, 90%, and 100% ethanol solutions). Here, samples were soaked for ten minutes at each step and soaked in 100% ethanol solution two times. Samples were then embedded in Epon812 epoxy resin before incubation. And three sequential curing phases were generated at 37 °C, 45 °C, and 65 °C, each for 24 h. Then, samples were sectioned using an UltracutE ultrathin slicer (Guangzhou Lingtuo Trading Co., Ltd., Guangzhou, China). Finally, they were stained with lead uranyl acetate and photographed using a JEM-1200EX transmission electron microscope (JEOL Corporation of Japan, Tokyo, Japan) from JEOL, Japan [46].

### 4.7. Methods for Microbial Enumeration in Apples

To assess microbial populations in apples, distinct protocols are employed for key pathogens and total colonies. *Penicillium* spp. are quantified using dichloran rose Bengal chloramphenicol agar (DRBC), which suppresses bacterial growth. Surface-disinfected apple tissues are homogenized, serially diluted, and plated, followed by incubation at 25 °C for 5–7 days; blue-green colonies with brush-like conidiophores confirm identification. *Aspergillus* spp. are cultured on malt extract agar (MEA) with chloramphenicol via pour-plating, incubated at 28 °C for 3–5 days, and identified by velvety black/brown colonies and radiate conidial structures. *Alternaria* spp. analysis involves potato dextrose agar (PDA) amended with streptomycin, with 7–10 days of incubation at 25 °C yielding dark olive-green colonies characterized by muriform conidia. For *Erwinia carotovora* (syn. *Pectobacterium carotovorum*), crystal violet pectate (CVP) medium selects pectinolytic bacteria; cavities around colonies after 24–48 h at 30 °C indicate pectin degradation, confirmed via Gram staining. Total aerobic mesophilic colonies are determined using plate count agar (PCA), with pour-plated dilutions incubated at 30 °C for 48–72 h and results expressed as CFU/g. Controls and molecular methods (e.g., PCR for *Erwinia pel* genes) ensure accuracy [47].

### 4.8. Volatile Extraction and Concentration

The determination of volatile, esters, alcohols, and aldehyde was performed with reference to Wang et al. [48]. In this investigation, SPME fibers (65 μm PDMS/DVB) were employed for the extraction of fragrance components based on the literature. Three grams of frozen material, three milliliters of NaCl (0.36 g/mL), and five microliters of the internal standard (0.0164 g/L 3-nonanone methanolic solution) were put into a twenty-milliliter sealed SPME vial for every extraction. Each treatment involved taking three fresh-cut apple slices for volatile extraction and concentration, with three repetitions per measurement. To ensure that all potential residues from the fiber coating were removed, the SPME fiber was first processed in the front GC injector for 30 min at 250 °C. After that, a Thermo RSH autosampler (Beijing Zeping Biotechnology Co., Ltd., Beijing, China) was used to remove the contents from the prepared sample vials. After 20 min of stirring at 40 °C, the combination equilibrated in both solution and headspace. After that, the fiber spent 20 min in the sealed SPME vial’s headspace. Ultimately, the fiber was removed and placed into the gas chromatograph’s heated injector port, where it was allowed to desorb for five minutes at 250 °C in split mode. The fiber was refurbished in the GC injector once the injection was finished.

### 4.9. GC–MS Conditions and Metabolomics Detection

A triple quadrupole mass spectrometer (TSQ 9000, Thermo Fisher Scientific, Waltham, MA, USA) and a gas chromatograph (TRACE 1310, Thermo Fisher Scientific) were employed in the GC–MS study. A TG-5MS (30 m × 0.25 mm I.D., 0.25 μm film, Thermo Fisher Scientific) capillary column was used to separate compounds. Helium was employed as the carrier gas, with a flow rate of 1 mL/min. The temperature was first set at 50 °C and held for 8 min using the GC’s program. It was then increased to 140 °C at a rate of 5 °C per minute and held for 2 min. Finally, it was raised to 270 °C at a rate of 10 °C per minute and held for 1 min. A split ratio of 5:1 was used for injecting the extracted volatile chemicals into the front GC injector. The temperature of the ion source was 290 °C. With the electron impact (EI) ionization mode and a scan range of 33–370 amu, we collected mass spectra at 70 eV.

In metabolomics detection, for LC-MS/MS, a C18 column is employed with a gradient of 0.1% formic acid in water (A) and acetonitrile (B), coupled to ESI-MS in positive/negative modes (*m*/*z* 50–1500). GC-MS analysis involves derivatization (methoximation and silylation) followed by separation on a DB-5MS column with temperature ramping (60 °C to 310 °C) and EI-MS (*m*/*z* 50–600). Raw data are processed using tools like XCMS or MS-DIAL for peak alignment, normalized to internal standards (e.g., ribitol for GC-MS), and analyzed via PCA/PLS-DA in MetaboAnalyst. Metabolites with VIP > 1.0, *p* < 0.05, and ≥2-fold change are annotated by matching MS/MS spectra and retention indices to databases (HMDB, KEGG, NIST) and validated with standards. Quality control includes pooled QC samples (RSD < 30%) to ensure reproducibility. Key metabolic pathways (e.g., glycolysis, phenolic metabolism) are mapped via KEGG or PlantCyc [49].

### 4.10. Determination of Aroma-Metabolism-Related Genes

The method reported by Gao et al. was used with slight changes [50]. Each treatment involved taking three fresh-cut apple slices for determination, with three repetitions per measurement. Uniform cortical tissue was excised from apples using sterile tools. Samples were snap-frozen in liquid nitrogen and stored at −80 °C until analysis. Total RNA was extracted using the rapid universal plant RNA extraction kit (Beijing Huayueyang Biotechnology, Beijing, China). The reverse transcription was performed using an M-MuLV first-strand cDNA synthesis kit (Shanghai Shenggong Technology, Shanghai, China) with the coding sequence of the apple gene, and the primer 5.0 software was used to design the gene-specific primers (Table 6). A random primer using oligo (dT) for the cDNA synthesis was subjected to the qRT-PCR experiment under the procedure using the ULtraSYBR Mixture (Low Rox) kit (Kangwei Century, Jiangsu, China).

### 4.11. Data Processing and Analysis

The statistical analysis followed rigorous validation protocols. Prior to conducting one-way ANOVA (SPSS 20.0), all datasets met ANOVA assumptions (*p* > 0.05 for both normality and variance homogeneity). Post hoc comparisons were performed using Duncan’s multiple-range test (α = 0.05), with results expressed as mean ± SD from triplicate independent experiments. Statistical significance was defined as *p* < 0.05.

## 5. Conclusions

Sorbitol-chelated calcium treatment enhances the postharvest quality of fresh-cut apples by synergistically stabilizing cellular structure and modulating metabolic activity. It mitigates enzymatic browning and microbial proliferation through calcium-mediated reinforcement of cell walls, which reduce oxidative damage and nutrient leakage. Concurrently, the treatment elevates aroma-related volatiles and stress-responsive metabolites by upregulating key biosynthetic genes, while preserving ultrastructural integrity and delaying tissue softening (Figure 8). This integrated approach effectively extends shelf life, maintains sensory attributes, and suppresses spoilage, positioning it as a holistic preservation strategy for fresh-cut produce.

## Figures and Tables

**Figure 1 plants-14-01293-f001:**
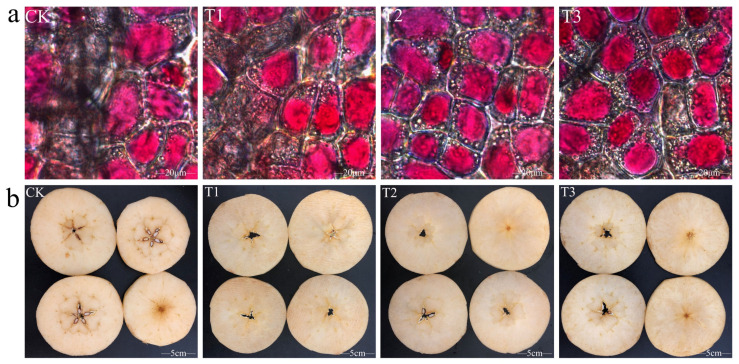
Changes in color difference after fresh-cutting of apples in different calcium preparation treatment groups. (**a**) Epidermal cells of Starkrimson apples, bars correspond to 20 μm. (**b**) The degree of browning of Starkrimson apples, bars correspond to 5 cm.

**Figure 2 plants-14-01293-f002:**
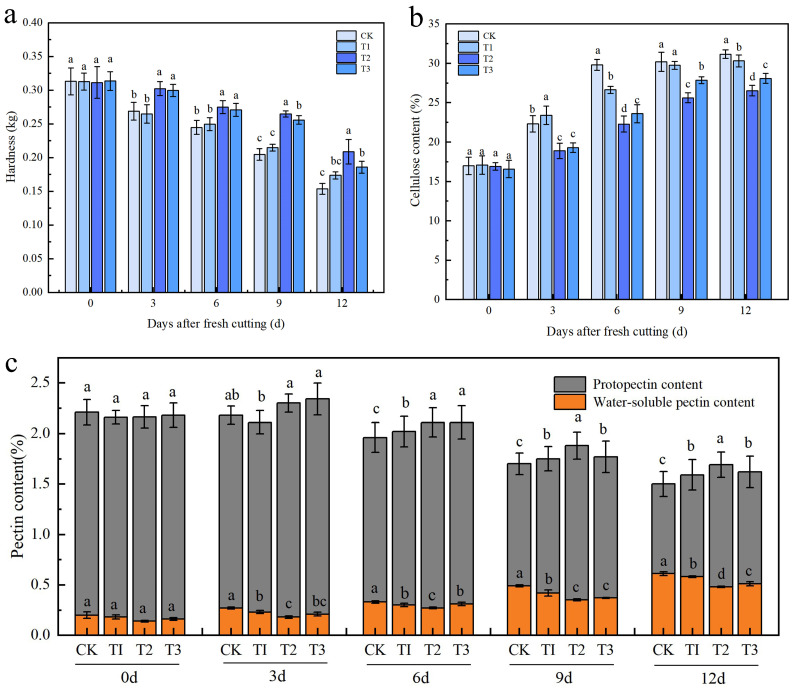
Changes in fruit hardness, cellulose content, and pectin contents after fresh-cutting of apples in different calcium preparation treatment groups. (**a**) Hardness of fresh-cut apples. (**b**) Cellulose content of fresh-cut apples. (**c**) Pectin contents of fresh-cut apples. The error bars represent the SD of three biological replicates. Different lowercase letters in the same developmental stage are significant differences at *p* < 0.05.

**Figure 3 plants-14-01293-f003:**
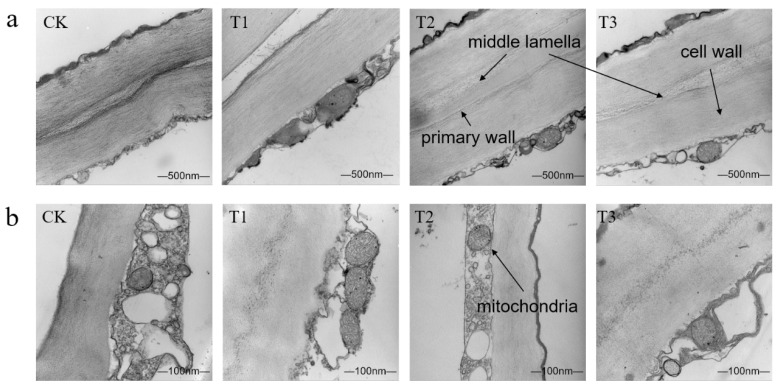
Changes in ultrastructure of apples in different calcium preparation treatment groups on the 12th day after fresh-cutting. (**a**) Bars correspond to 500 nm. (**b**) Bars correspond to 100 nm. The black arrows in the figures are middle lamella, primary wall, mitochondria, and cell wall.

**Figure 4 plants-14-01293-f004:**
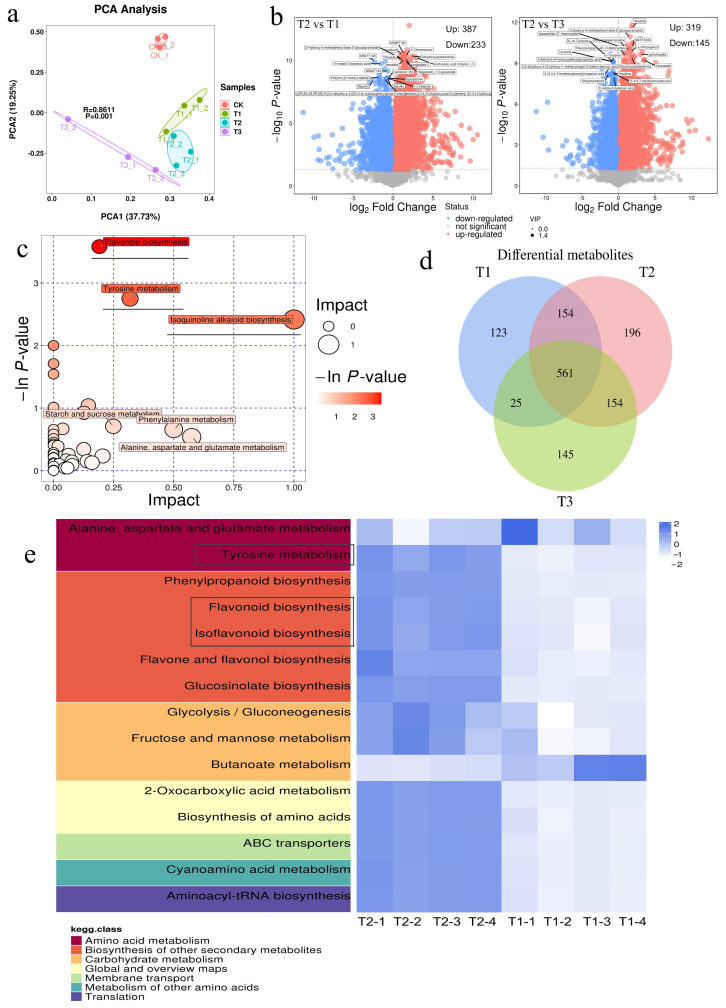
Effects of sorbitol-chelated calcium on metabolomics of fresh-cut apples during storage. (**a**) Principal component analysis, PCA plot. (**b**) Volcano plot of differential metabolites. (**c**) Pathway analysis for T2 vs. T1 groups. (**d**) Venn diagram of differential metabolites. (**e**) KEGG heatmap for T2 vs. T1 groups.

**Figure 5 plants-14-01293-f005:**
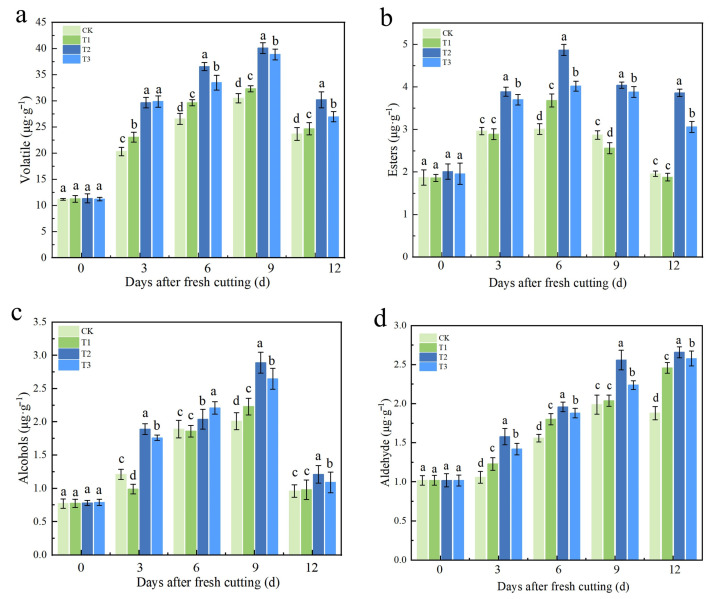
Effects of different calcium treatments after harvest on aroma and volatile components of fruit during storage. (**a**) Content of total volatiles. (**b**) Content of ester substances. (**c**) Content of alcohols. (**d**) Content of aldehydes. CK, T1, T2, and T3 represent the different treatments of control, calcium chloride, sorbitol-chelated calcium, and calcium nitrate. The error bars represent the SD of three biological replicates. Different lowercase letters in the same developmental stage indicate a significant difference at *p* < 0.05.

**Figure 6 plants-14-01293-f006:**
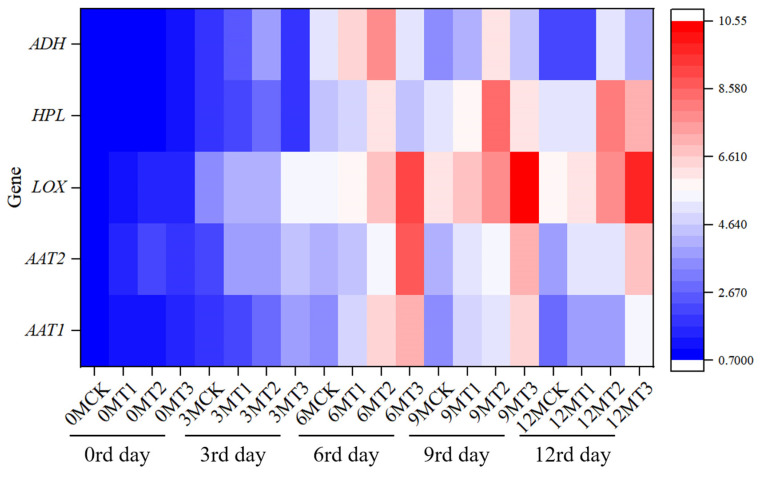
Effects of different calcium preparations after harvesting on the expression of volatiles-related genes in apples during storage. CK, T1, T2, and T3 represent the different treatments of control, calcium chloride, sorbitol-chelated calcium, and calcium nitrate. The 0MCK represents the 0th day CK group, 3MCK represents the 3rd day CK group, and so on.

**Figure 7 plants-14-01293-f007:**
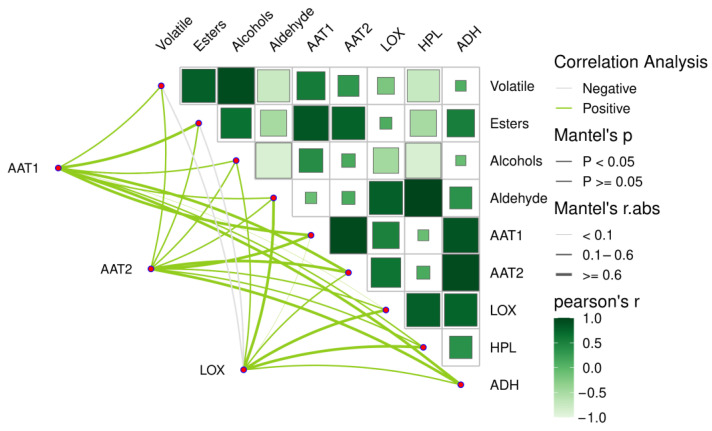
Heatmap of correlation analysis between volatile matter and related gene expression in apple during storage.

**Figure 8 plants-14-01293-f008:**
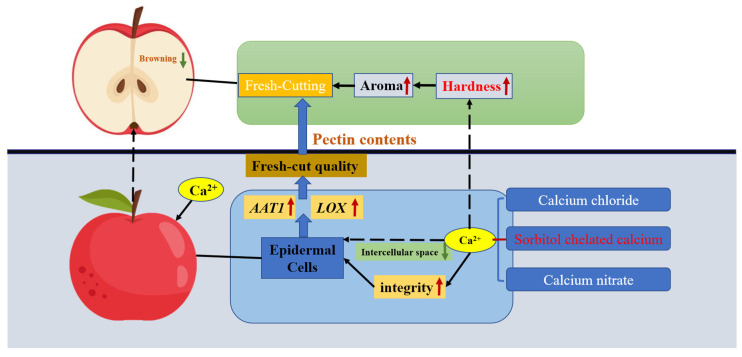
Graphical abstract of the different calcium preparations for fresh-cut quality and storage quality of Starkrimson apple.

**Table 1 plants-14-01293-t001:** Effect of different calcium preparation groups on the change in color value of fresh-cut apple (a* (red/green), b* (blue/yellow), L* (lightness)) during storage.

Color Hues	Treatment	Days After Fresh-Cutting (d)				
		0 d	3 d	6 d	9 d	12 d
**a* value**	CK	−5.59 ± 0.31 ^a^	−4.91 ± 0.18 ^c^	−4.62 ± 0.34 ^c^	−4.47 ± 0.27 ^c^	−4.29 ± 0.21 ^c^
	T1	−5.41 ± 0.31 ^a^	−4.57 ± 0.23 ^b^	−4.33 ± 0.35 ^b^	−4.2 ± 0.25 ^b^	−3.42 ± 0.23 ^b^
	T2	−5.37 ± 0.38 ^a^	−4.23 ± 0.26 ^a^	−3.37 ± 0.42 ^a^	−2.96 ± 0.10 ^a^	−2.90 ± 0.15 ^a^
	T3	−5.43 ± 0.23 ^a^	−4.54 ± 0.25 ^b^	−4.35 ± 0.23 ^b^	−4.25 ± 0.39 ^b^	−3.45 ± 0.17 ^b^
**b* value**	CK	26.91 ± 0.32 ^a^	25.67 ± 0.52 ^b^	25.34 ± 0.55 ^b^	24.76 ± 0.62 ^b^	23.87 ± 0.62 ^c^
	T1	27.01 ± 0.24 ^a^	26.29 ± 0.43 ^a^	25.34 ± 0.32 ^b^	24.86 ± 0.55 ^b^	24.25 ± 0.32 ^b^
	T2	27.21 ± 0.85 ^a^	26.44 ± 0.22 ^a^	26.17 ± 0.22 ^a^	25.59 ± 0.13 ^a^	25.34 ± 0.23 ^a^
	T3	26.91 ± 0.12 ^a^	25.34 ± 0.27 ^b^	24.49 ± 0.27 ^b^	24.69 ± 0.81 ^b^	23.86 ± 0.14 ^c^
**L* value**	CK	79.01 ± 1.08 ^a^	73.54 ± 0.75 ^c^	69.7 ± 0.66 ^d^	68.28 ± 0.51 ^c^	65.05 ± 0.55 ^d^
	T1	79.37 ± 0.66 ^a^	74.03 ± 0.66 ^b^	71.85 ± 0.42 ^c^	70.47 ± 0.84 ^b^	66.81 ± 0.61 ^c^
	T2	79.78 ± 0.61 ^a^	79.18 ± 0.83 ^a^	78.40 ± 0.93 ^a^	77.48 ± 0.82 ^a^	72.03 ± 0.98 ^a^
	T3	78.58 ± 0.67 ^a^	75.17 ± 0.55 ^b^	73.71 ± 0.61 ^b^	71.03 ± 0.64 ^b^	68.12 ± 0.65 ^b^

Note: Different letters indicate a significant difference between treatments at each time point (*p* < 0.05) by Duncan’s multiple range (DMR) test.

**Table 2 plants-14-01293-t002:** Changes in spoilage microorganisms on the 12th day after fresh-cutting of apples in different calcium preparation treatment groups (lg CFU/g^−1^).

Microbial Species	Different Calcium Preparation Treatment Groups			
	CK	T1	T2	T3
*Penicillium*	2.34 ± 0.09 ^a^	1.89 ± 0.08 ^b^	1.13 ± 0.09 ^d^	1.77 ± 0.02 ^c^
*Aspergillus*	1.43 ± 0.06 ^a^	1.33 ± 0.06 ^b^	1.07 ± 0.06 ^c^	1.12 ± 0.04 ^bc^
*Alternaria*	2.89± 0.11 ^a^	1.87 ± 0.13 ^b^	1.61 ± 0.07 ^c^	1.76 ± 0.04 ^b^
*Erwinia carotovora*	4.43 ± 0.12 ^a^	3.34 ± 0.13 ^b^	2.97 ± 0.15 ^c^	3.03 ± 0.12 ^c^
Total colonies	5.68 ± 0.18 ^a^	4.80 ± 0.12 ^b^	3.70 ± 0.09 ^d^	4.19 ± 0.13 ^c^

Note: Different lowercase letters in the same developmental stage are significant differences at *p* < 0.05.

**Table 3 plants-14-01293-t003:** Volatile compounds in freshly cut apples on day 12.

Volatile Compound	Retention Index	Markers	Mass Concentration (mg·kg^−1^)			
			CK	T1	T2	T3
Ethanol	1.66		0.01 ± 0.001 ^c^	0.02 ± 0.001 ^b^	0.02 ± 0.002 ^b^	1.59 ± 0.311 ^a^
Silanol, trimethyl-	1.792	V	1.51 ± 0.123 ^a^	0.78 ± 0.074 ^c^	1.2 ± 0.083 ^b^	0.3 ± 0.023 ^d^
Pentanoic acid, 3-methyl-4-oxo-	1.996	V	0.35 ± 0.025 ^b^	0.92 ± 0.08 ^a^	0.36 ± 0.02 ^b^	0.22 ± 0.014 ^c^
Silanediol, dimethyl-	2.335	V	0.26 ± 0.019 ^b^	0.12 ± 0.006 ^d^	0.19 ± 0.012 ^c^	0.62 ± 0.044 ^a^
Hexanal	3.339	V	1.49 ± 0.081 ^c^	0.27 ± 0.019 ^b^	0.38 ± 0.025 ^b^	5.66 ± 0.471 ^a^
Cyclotrisiloxane, hexamethyl-	5.136	V	7.39 ± 0.548 ^a^	0.77 ± 0.073 ^b^	0.18 ± 0.009 ^d^	0.43 ± 0.027 ^c^
kaempferol-	5.606	V	0.03 ± 0.002 ^d^	3.94 ± 0.27 ^b^	5.04 ± 0.436 ^a^	0.82 ± 0.041 ^c^
2-Hexenal	5.817	V	0.03 ± 0.002 ^c^	0.26 ± 0.015 ^b^	0.41 ± 0.031 ^a^	0 ± 0 ^d^
2-Hexenal	6.333	V	0.01 ± 0.001 ^c^	0.91 ± 0.047 ^a^	0.73 ± 0.055 ^b^	0.01 ± 0.001 ^c^
4-Hexen-1-ol, acetate	6.454	V	0.25 ± 0.015 ^a^	0.15 ± 0.012 ^b^	0.02 ± 0.002 ^c^	0.24 ± 0.016 ^a^
2-Hexen-1-ol, (E)-	6.543		1.15 ± 0.096 ^a^	0.1 ± 0.007 ^d^	0.69 ± 0.052 ^b^	0.16 ± 0.009 ^c^
Cyclopropane, propyl-	6.779		0.1 ± 0.006 ^d^	0.45 ± 0.031 ^b^	0.24 ± 0.019 ^c^	0.76 ± 0.044 ^a^
1-Butanol, 3-methyl-, acetate	6.817	V	0.05 ± 0.003 ^d^	0.18 ± 0.009 ^c^	1.1 ± 0.056 ^a^	0.83 ± 0.061 ^b^
Oxime-, methoxy-phenyl-_	7.01	SV	0.28 ± 0.026 ^b^	0.84 ± 0.044 ^a^	0.98 ± 0.084 ^a^	0.13 ± 0.012 ^c^
Acetic acid, pentyl ester	7.838	T	0.14 ± 0.007 ^b^	0.15 ± 0.009 ^b^	0.12 ± 0.008 ^c^	0.64 ± 0.054 ^a^
2-Hexen-1-ol, acetate, (Z)-	8.016	V	0.78 ± 0.067 ^a^	0.04 ± 0.003 ^c^	0.02 ± 0.001 ^c^	0.14 ± 0.007 ^b^
4′,6′-Dimethoxy-2′,3′-dimethylacetop	8.21	V	0.14 ± 0.008 ^b^	0.11 ± 0.006 ^b^	0.7 ± 0.05 ^a^	0.03 ± 0.003 ^c^
3-Hydroxy-4-methoxybenzaldehyde	9.106	V	0.25 ± 0.013 ^a^	0.14 ± 0.011 ^c^	0.2 ± 0.014 ^b^	0.06 ± 0.004 ^d^
Cyclotrisiloxane, hexamethyl-	9.215	V	0.02 ± 0.001 ^c^	0.01 ± 0.001 ^d^	0.07 ± 0.004 ^b^	0.17 ± 0.011 ^a^
5-Hepten-2-one, 6-methyl-	9.833	V	0.15 ± 0.014 ^a^	0.02 ± 0.002 ^c^	0.05 ± 0.004 ^b^	0.01 ± 0.001^c^
2-Octanone	9.954		1.98 ± 0.127 ^a^	0.13 ± 0.011 ^c^	0.18 ± 0.011 ^b^	0.16 ± 0.011 ^b^
Cyclotetrasiloxane, octamethyl-	10.029	V	1.02 ± 0.055 ^c^	2.22 ± 0.117 ^a^	0.01 ± 0.001 ^d^	1.88 ± 0.183 ^b^
2-Octanol, (S)-	10.215	SV	15.29 ± 0.882 ^a^	0.72 ± 0.06 ^b^	0.06 ± 0.003 ^c^	0.59 ± 0.057 ^b^
3-Hexen-1-ol, acetate, (E)-	10.335	T	11.39 ± 0.765 ^a^	10.69 ± 0.568 ^b^	1.61 ± 0.138 ^c^	12.45 ± 0.72 ^a^
Acetic acid, hexyl ester	10.48	S	15.31 ± 1.205 ^a^	12.91 ± 1.26 ^b^	0.45 ± 0.032 ^c^	16.89 ± 0.951 ^a^
2-Hexen-1-ol, acetate, (Z)-	10.664	T	17.64 ± 0.965 ^b^	21.05 ± 1.263 ^a^	2.35 ± 0.143 ^c^	21.6 ± 1.153 ^a^
Nonane, 5-butyl-	10.758		0.02 ± 0.001 ^c^	24.24 ± 1.268 ^a^	10.17 ± 0.705 ^b^	23.47 ± 1.389 ^a^
Quercetin	11.575		0.01 ± 0.001 ^b^	0.02 ± 0.001 ^b^	16.32 ± 0.867 ^a^	0.01 ± 0.001 ^b^
Decane, 3,7-dimethyl-	11.683		0.13 ± 0.008 ^b^	0.01 ± 0.001 ^c^	22.2 ± 1.737 ^a^	0.01 ± 0.001 ^c^
hydroxytyrosol-	11.761	SV	0.06 ± 0.003 ^c^	0.25 ± 0.024 ^b^	24.04 ± 1.202 ^a^	0.24 ± 0.014 ^b^
Nonane, 5-(2-methylpropyl)-	11.842	T	0.02 ± 0.002 ^a^	0.02 ± 0.001 ^a^	0.01 ± 0.001 ^b^	0.02 ± 0.001 ^a^
4H-3,1-Benzoxazin-2-amine, 4-ethyl-N-	11.904	V	0.92 ± 0.087 ^a^	0.1 ± 0.007 ^b^	0.01 ± 0.001 ^c^	0.11 ± 0.009 ^b^
1-Hexene, 3,3,5-trimethyl-	12.142	V	0.17 ± 0.009 ^b^	0.02 ± 0.002 ^c^	0.24 ± 0.013 ^a^	0.02 ± 0.001 ^c^
3-Nonanone	12.333	V	0.12 ± 0.007 ^a^	0.03 ± 0.002 ^b^	0.02 ± 0.002 ^c^	0.01 ± 0.001 ^d^
Octane, 2,3,6,7-tetramethyl-	12.592	V	0.36 ± 0.032 ^a^	0.32 ± 0.021 ^a^	0.11 ± 0.006 ^b^	0.01 ± 0.001 ^c^
Nonane, 2-methyl-	12.739	V	0.27 ± 0.017 ^a^	0.15 ± 0.009 ^b^	0.01 ± 0.001 ^c^	0.03 ± 0.002 ^c^
Dodecane, 2,6,11-trimethyl-	12.867	V	0.07 ± 0.004 ^b^	0.09 ± 0.005 ^a^	0.02 ± 0.002 ^d^	0.04 ± 0.003 ^c^
Nonane, 4,5-dimethyl-	12.921	V	0.78 ± 0.041 ^a^	0.29 ± 0.028 ^b^	0.03 ± 0.003 ^c^	0.27 ± 0.017 ^b^
Tyrosol-	13.06		0.03 ± 0.002 ^c^	0.01 ± 0.001 ^c^	0.3 ± 0.027 ^a^	0.17 ± 0.012 ^b^
5,6,6,7-Tetramethyl-13-oxa-5,7-	13.125		0.02 ± 0.002 ^c^	0.33 ± 0.018 ^a^	0.16 ± 0.008 ^b^	0.07 ± 0.006 ^c^
trans-2-Heptenyl acetate	13.232	V	0.14 ± 0.008 ^b^	0.07 ± 0.004 ^c^	0.06 ± 0.003 ^c^	0.27 ± 0.026 ^a^
2-Octanol, acetate	13.333	V	0.03 ± 0.002 ^d^	0.71 ± 0.045 ^a^	0.28 ± 0.017 ^b^	0.07 ± 0.007 ^c^
Phosphonoacetic acid, 3TMS derivative	13.89	V	0.05 ± 0.003 ^b^	0.01 ± 0.001 ^b^	0.1 ± 0.001 ^b^	0.23 ± 0.018 ^a^
Cyclopentasiloxane, decamethyl-	14.054	V	0.01 ± 0.001 ^c^	0.04 ± 0.003 ^b^	0.26 ± 0.015 ^a^	0.06 ± 0.005 ^b^
Undecane, 3,4-dimethyl-	14.225	V	0.17 ± 0.015 ^b^	0.01 ± 0.001 ^d^	0.05 ± 0.003 ^c^	0.62 ± 0.044 ^a^
Isoflavones	14.475		0.04 ± 0.002 ^b^	0.02 ± 0.001 ^c^	0.57 ± 0.034 ^a^	0.01 ± 0.001 ^c^
Pentadecane	15.374	V	0.05 ± 0.003 ^a^	0.01 ± 0.001 ^b^	0.01 ± 0.001 ^b^	0.04 ± 0.002 ^a^
3-Isopropoxy-1,1,1,5,5,5-hexamethyl-3-	15.483	V	0.05 ± 0.003 ^a^	0.03 ± 0.002 ^b^	0.05 ± 0.003 ^a^	0.01 ± 0.001 ^c^
Decanal	15.549	V	0.05 ± 0.003 ^a^	0.02 ± 0.001 ^b^	0.02 ± 0.002 ^b^	0.02 ± 0.001 ^b^
Undecane, 4,6-dimethyl-	15.625	V	0.31 ± 0.022 ^a^	0.01 ± 0.001 ^d^	0.02 ± 0.001 ^c^	0.06 ± 0.004 ^b^
Dodecane, 4-methyl-	15.692	V	0.01 ± 0.001 ^c^	0.01 ± 0.001 ^c^	0.02 ± 0.001 ^b^	0.05 ± 0.003 ^a^
Decane, 2,4,6-trimethyl-	15.879	V	0.86 ± 0.062 ^a^	0.12 ± 0.007 ^b^	0.03 ± 0.002 ^c^	0.02 ± 0.001 ^c^

Note: Different lowercase letters in the same developmental stage are significant differences at *p* < 0.05.

**Table 4 plants-14-01293-t004:** The correlation between aroma substances and related genes.

	Volatile	Esters	Alcohols	Aldehyde	*AAT1*	*AAT2*	*LOX*	*HPL*	*ADH*
Volatile	1								
Esters	0.884 *	1							
Alcohols	0.909 *	0.735	1						
Aldehyde	0.588	0.554	0.297	1					
*AAT1*	0.898 *	0.921 *	0.777	0.740	1				
*AAT2*	0.825	0.934 *	0.672	0.736	0.983 **	1			
*LOX*	0.874	0.830	0.671	0.895 *	0.950 *	0.921 *	1		
*HPL*	0.605	0.555	0.341	0.997 **	0.762	0.750	0.906 *	1	
*ADH*	0.783	0.848	0.668	0.770	0.976 **	0.978 **	0.923 *	0.797	1

Note: * The significant level, *p* < 0.05; ** The significant level, *p* < 0.01.

**Table 5 plants-14-01293-t005:** Experiment design.

Treatment	Control	Calcium Chloride	Sorbitol-Chelated Calcium	Calcium Nitrate
CK	Water	—	—	—
T1	—	4%	—	—
T2	—	—	4%	—
T3	—	—	—	4%

**Table 6 plants-14-01293-t006:** Primers for the qRT-PCR amplification of aroma-substance-related gene expression.

Gene Name	Forward Sequence of the Primers (5′–3′)	Reverse Sequence of the Primers (5′–3′)
*AAT1*	GCTGGATTGCTCTTGTTC	TGGTTACTGGATGCGTAT
*AAT2*	GGATTACTCAGGAACCTAA	GACACAACTCTACATTGC
*LOX*	GATGGTCTCCTCGTATGG	CTTCGTGTCCCTTATTCTTG
*ADH*	CCACCACAAGCAAATGAA	ACCAACACTCTCCACAAT
*HPL*	TAGGAGGGAAGTGAGAGG	AGAGAAACAAAGCGAGGT
Actin	TGACCGAATGAGCAAGGAAATTACT	TACTCAGCTTTGGCAATCCACATC

## Data Availability

The data presented in this study are not available due to privacy.

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
