# Peer review of "Effects of Different Calcium Preparations on Fresh-Cut Quality and Storage Quality of Starkrimson Apple"

_plants, 2025, doi:10.3390/plants14091293_

Round 1

Reviewer 1 Report

Comments and Suggestions for Authors

The comments are in the attached file

Reviewer 2 Report

Comments and Suggestions for Authors

The manuscript entitled “Effects of sorbitol-chelated calcium preparations on fresh-cut quality and storage quality of Starkrimson apple” reports that sorbitol-chelated calcium maintained fruit texture and pectin degradation, enhanced volatile synthesis and reduced microbial spoilage and browning during storage. This work is valuable to improve the storage quality of fresh-cut apples and the study is well-presented. However, some figures and tables should be changed in order to improve the readability of the study. Additionally, the ambiguous statements should be clarified by the authors. Thank you.

Abstract

Line 19-21: Other treatments (T1 and T3) should be introduced in brief.

Line 24: Delete T2. The treatment name has described already.

Result

Table 1. The results and statistical analysis are difficult to follow. The results are compared with treatments (CK, T1, T2, T3) on each time point? If compare within treatments, a* value on 6 d is wrong. Different letters indicated CK and T1 are “c” and T2 and T3 are “a”. Where is “b” for that time point? The results and table should be checked and changed to figure.

Figure 2C. What do you mean original pectin content? Do you mean total pectin?

Figure 2C. Changing the treatment name from CK, T1, T2, T3 to A, B, C, D makes complexity to the readers. Just use the treatment names (CK, T1, T2, T3) and add 0 day, 3 days,…… under the treatment name. It is not difficult and just use a sample way.

How about other fruit quality indicators? The authors explained that sorbitol is a sugar alcohol in the introduction, and why the results of soluble solids sugars and titratable acidity are not presented?

Since calcium accumulation is fruit tissue is linked in improving cell wall structural and integrity, the amount of calcium concentration in fruit after different calcium treatments should be analyzed and presented in order to provide a comprehensive understanding in the role of calcium in cell wall degradation.

Reviewer 3 Report

Comments and Suggestions for Authors

The manuscript entitled "Effects of sorbitol-chelated calcium preparations on fresh-cut quality and storage quality of Starkrimson apple" needs major revision.

Since different calcium sources were used, the title should be revised to reflect this.

Please explain why it is necessary to conduct research on fresh-cut apples.

What is the rationale behind selecting these specific calcium sources?

The discussion section also needs improvement.

Round 2

Reviewer 2 Report

Comments and Suggestions for Authors

The authors addressed all my comments and revised the manuscript properly.

I suggest that calcium accumulation in fruits should be measured in future researches.

Thank you. 

Reviewer 3 Report

Comments and Suggestions for Authors

Accept in present form